# An Investigation into the Performance and Mechanisms of Soymilk-Sized Handmade Xuan Paper at Different Concentrations of Soymilk

**DOI:** 10.3390/molecules28196791

**Published:** 2023-09-25

**Authors:** Chunfang Wu, Yangyang Liu, Yanxiao Hu, Ming Ding, Xiang Cui, Yixin Liu, Peng Liu, Hongbin Zhang, Yuliang Yang, Hongdong Zhang

**Affiliations:** 1Institute for Preservation and Conservation of Chinese Ancient Books, Fudan University, Shanghai 200433, China; 19110820002@fudan.edu.cn (C.W.);; 2School of Creative Art and Fashion Design, Huzhou Vocational and Technical College, Huzhou 313000, China; 3Behavioral and Cognitive Neuroscience Center, Institute of Science and Technology for Brain-Inspired Intelligence, Fudan University, Shanghai 200433, China; 4Department of Rehabilitation Medicine, Huashan Hospital, Fudan University, Shanghai 200040, China; 5State Key Laboratory of Molecular Engineering of Polymers, Department of Macromolecular Science, Fudan University, Shanghai 200433, China

**Keywords:** surface sizing, soymilk, Xuan paper, lifespan, potential mechanism

## Abstract

Invaluable paper relics that embody a rich traditional culture have suffered damage, requiring urgent restoration. In this context, the utilization of soymilk as a sizing agent holds great significance and reverence. This study investigates the use of soymilk as a sizing agent for Xuan paper and evaluates its effects on various properties and the long-term behavior of the paper. The findings reveal that the application of soymilk as a sizing agent for Xuan paper imparts distinct properties, including hydrophobicity, improved mechanical properties, and unique chromaticity. These characteristics—arising from the papillae on the surface of the Xuan paper, the protein folding of the soy protein, and hydrogen-bonding interactions between the soy protein and paper fibers—play a crucial role in shaping the paper’s unique attributes. From a physicochemical perspective, the aging process leads to multiple changes in paper properties. These changes include acidification, which refers to a decrease in pH, as well as a decline in mechanical strength, an increase in chromaticity, and a decrease in the degree of polymerization (DP) of the paper. The Ekenstam equation is employed to predict the lifespan of the paper, showing longer lifespans for Sheng Xuan paper and a negative correlation between soymilk concentration and lifespan in soymilk-sized paper. Our work provides valuable insights for the preservation and maintenance of paper, highlighting the potential benefits and challenges of using soymilk for surface sizing.

## 1. Introduction

Traditional Chinese calligraphy and painting have always been an important carrier for Chinese literati to express personal emotions or concerns about political life, reflect on social changes, and record historical events, and it is an integral part of the outstanding traditional Chinese culture. From 2012 to 2016, the first national survey of movable cultural relics was carried out under the unified deployment of The State Council of the People’s Republic of China. Census data have shown that there are more than 1.5 million paintings and calligraphy cultural relics in China [1]. However, these cultural relics have suffered varying degrees of damage, either due to environmental factors such as armed conflict, conflagration, seismic activity, floods, insolation, and insect infestation, or as a result of human actions like frequent flipping and tearing over time. In order to prevent further damage, most of these cultural relics are facing the urgent need of restoration. Furthermore, the restoration of traditional Chinese paintings and calligraphy has consistently upheld the principle of “Repairing the old as the old” [2]. This principle entails meticulously repairing the damaged sections of cultural artifacts while aiming to restore the original aesthetics of the artwork, ensuring that no visible traces of restoration remain discernible to observers.

During the restoration process of traditional paintings and calligraphy, starch is commonly used as a natural sizing agent. However, natural starch itself has certain limitations, including the high viscosity of starch paste, a poor affinity with fibers, and a tendency to detach easily [3]. These limitations restrict the practical application of natural starch to some extent. In response to these challenges, our ancestors discovered and refined a valuable technique known as the mixed use of alum and gelatin through continuous practice and exploration. When employed in the restoration of calligraphy and paintings, an alum–gelatin solution can effectively secure the pigment on the artwork, preventing it from flaking off. In addition, it enhances the water resistance of the paper. However, it is crucial to find the appropriate ratio of alum to gelatin. If the proportion of alum is too high, it can render the paper cultural relics brittle. The use of an alum–gelatin solution in the restoration of paper cultural relics has been a topic of debate, as there are concerns about its potential harm [4]. Therefore, the demand for safe and harmless sizing agents in cultural relic restoration remains unmet. Soybean, one of the most important global crops, is widely cultivated and consumed worldwide. It contains a multitude of proteins, lipids, and secondary metabolites, granting it a crucial position across various sectors [5]. Hereinto, soybean protein exhibits amphiphilic properties, which have garnered considerable attention in fields such as medicine, beauty, packaging, and food. In particular, soybean protein has also captured significant interest within the domain of cultural relics conservation due to its amphiphilic nature. In the realm of traditional Chinese painting and calligraphy restoration, soybean-derived soymilk has long been utilized as a sizing agent. Esteemed expert Ms. Qiu Jinxian, renowned for her exceptional proficiency in restoring valuable paintings and calligraphy, has contributed significantly to this field during her tenure at both the Shanghai Museum and the British Museum. In her report, Ms. Qiu highlighted that soymilk exhibited remarkable suitability in the restoration of calligraphy and paintings, possessing favorable color properties and hydrophilic and hydrophobic performance. Typically, the recommended concentration is one cup of soymilk with three cups of water [6]. The hydrophilic and hydrophobic properties of soymilk-sized paper can be traced back to the Ming Dynasty. The eighth volume of *Physical Knowledge* written by Fang Yizhi, a Ming Dynasty scholar, documents the process of transforming paper into waterproof window paper with ‘‘tofu pulp’’. In addition, the painter Wu Shanming suggested that the ideal ratio of soymilk to water should be maintained at three to seven [7], which yields the optimal hydrophilic and hydrophobic performance when sizing paper with soymilk.

According to Xu’s research, the application of soybean protein as a coating on paper surfaces reduces surface voids and facilitates the formation of hydrogen bonds between protein and fiber hydroxyl groups. This interaction enhances the water resistance of the paper, making soybean protein an effective sizing agent for improving its hydrophobic property [8]. In addition, when soymilk is used as a sizing agent at different temperatures, it is observed that the hydrophilic and hydrophobic performance of the paper can be varied. Specifically, the soybean protein molecules gradually unfold from their tightly coiled structure, which is surrounded by a hydration film, and remain relatively stable. This unfolding exposes the hydrophobic groups within the polypeptide chain’s coiled structure while reducing the presence of hydrophilic groups on the outer surface [9]. As the temperature of the soymilk increases from 25 °C to 80 °C, the relative content of the β-sheet structure (one of the secondary structures of protein molecules) increases from 41.14% to 48.87%. Meanwhile, the corresponding α-helix, β-turn, and random coiled structures of soybean protein transform into β-sheet structures, with the proportion of β-sheet structures positively correlated with hydrophobicity. However, the relative content of the β-sheet structure in soybean protein decreases to 45.43% at 90 °C. This reduction in β-sheet structure content indicates that the structure of soybean protein can undergo thermal denaturation at higher temperatures [10]. Consequently, precipitation may occur, compromising the desired properties of the soybean protein sizing agent.

The use of soymilk as a sizing agent for paper either lacks specific information, or the information provides vague details regarding the appropriate ratio of soymilk to water. This study focuses on the influence of the varying concentrations of soymilk on the paper when it is used as a sizing agent. A total of six distinct types of paper were carefully chosen to conduct a comprehensive analysis of the surface properties and the impact of soymilk treatment at both macro and micro levels. In addition, this study also evaluated the lifespan of paper that treated with soymilk. The findings of this research will aid in the selection of optimal soymilk concentrations for paper sizing, thereby establishing a strong foundation for understanding the interaction between paper fibers and soybean protein.

## 2. Results and Discussion

### 2.1. Characterization of Soymilk

In Figure 1A, the pH values of soymilk at different concentrations are close to neutral, indicating that they do not pose a threat to paper when used for sizing. The viscosity of soymilk is found to be closely related to its concentration. Higher concentrations of soymilk result in greater viscosity. Herein, it was observed that the 1:5 concentration of soymilk had the lowest viscosity, which was similar to the viscosity of water at ambient temperature (1.01 mPa·s). A lower viscosity indicates strong flowability and easy spreading of the soymilk. This characteristic is beneficial for the soymilk to penetrate into the paper [11]. Therefore, it can be concluded that the experimental concentration of soymilk allows for easy permeation of the paper due to its low viscosity and favorable flowability.

The main ingredient in soymilk is soy proteins, which are composed of albumins and globulins. When extracted by water, albumins represent approximately 10% of soy proteins. Soy protein molecules are complex macromolecules consisting of 20 different amino acids. These amino acids are connected through peptide bonds, forming polypeptide chains, which represent the primary structure of protein. The amino acids’ skeleton comprises hydrophilic amino groups, the carboxylic groups, and hydrophobic side chains. Each amino acid, with its specific side chain, plays a unique role in the protein structure [11]. The presence of polar groups and apolar groups in proteins influences the structure in an aqueous solution. Therefore, the surface charge and size distribution of the soy protein were measured under different pH conditions. Figure 1B shows that the zeta potential of soy protein gradually decreases with increasing pH, reaching its isoelectric point at pH 5.40. Many researchers have demonstrated that polysaccharide–protein composite films exhibit improved functional properties. The maximum complexation interaction between protein and polysaccharide, mediated by electrostatic interactions, typically occurs near the isoelectric point. Therefore, the surface potential of soymilk plays a role in its interaction with paper cellulose and can influence paper properties to some extent.

In addition, Figure 1C displays the size distribution of soymilk proteins, revealing that protein particles range in size from 45 nm to 8 μm. Previous research has shown that protein samples can exhibit multiple distributions under certain conditions. Some particles are distributed in the nanoscale range, representing natural protein molecules, while others exist as aggregates of protein molecules and are larger in size [12]. Specifically, as the pH decreases and the proton concentration increases, the amino and carboxylic groups in the protein interact with these protons, resulting in a positively charged surface and a low net charge. At pH 3.64, the net surface charge of protein is +1.82 mV. The repulsion between soy protein and protons leads to a primary protein size distribution ranging from 600 to 1100 nm. With the increase in pH, the proton concentration decreases, weakening the interaction. This change results in a wide range of protein sizes. At pH 4.53, protein sizes exhibit a distribution spanning between 600 to 2000 and 4000 to 8000 nm. As the pH approaches the isoelectric point, polydisperse particles become more noticeable due to reduced electrostatic repulsion and decreased solution stability. The size distribution spans two orders of magnitude at pH 5.43. This aggregation tendency is attributed to the presence of protein molecules with less homogeneous charges on their surfaces [13]. As the pH increases, some amino acids in the protein release protons, leading to a negatively charged environment. The reduction in proton concentration with increasing pH causes an increase in the net negative charge on the protein surface. Electrostatic repulsions between proteins can cause aggregation and a decrease in particle size under high pH conditions [14]. The protein size distribution decreased from 3000~8000 nm at pH 6.13 to 100~600 nm at pH 8.54. It has been reported that the free energy of protein has a strong negative correlation with protein size and microstructure, which, in turn, affects the functionalities [15].

### 2.2. Analysis of Sized Paper

The quality of paper can be evaluated by basic weight, which considers the basic weight parameter and the influences of physical and printing properties on the paper. In Table 1, the quality of paper coated with different sizing agents is presented. The basic weights of the soymilk-coated paper are higher compared with Sheng Xuan paper, indicating an improvement in quality. The basic weight of soymilk-sized paper decreases as the concentration of soymilk decreases. This means that lower concentrations of soymilk result in a decrease in the weight of the paper. In addition, the basic weight of the soymilk-sized paper is lower compared with paper coated with an alum–gelatin solution. The results indicate a distinct weight advantage of the soymilk-coated paper over Shu Xuan paper, suggesting that soymilk sizing can provide a lighter-weight paper option while maintaining or even improving quality.

Investigating the dynamic behavior of liquids on surfaces is crucial for understanding the surface properties of Xuan paper and its practical applications in writing and printing. Typically, a water-wetting threshold of 90° is considered to divide surfaces into hydrophilic and hydrophobic categories. Figure 2A shows that the static contact angle of the soymilk-sized paper is significantly improved compared with Sheng Xuan paper and is comparable to that of Shu Xuan. This indicates that the soymilk-sized paper possesses a more hydrophobic surface compared with the uncoated Sheng Xuan paper. Notably, the contact angles of the 3:7 and 1:3 soymilk-sized papers are larger than the other concentrations, demonstrating higher hydrophobic properties and water resistance. Restorers and painters often prefer using these concentrations of soymilk-sized paper in practical applications. Paper with low surface energy allows for controlled surface properties, such as producing a hydrophobic and self-cleaning coating. In Figure 2B, it can be observed that Shu Xuan paper, which exhibits the maximum contact angle, possesses the lowest surface energy. The smaller the water contact angle on the surface of the paper, the stronger the interaction between the water and the paper surface, indicating a higher surface energy. Therefore, Sheng Xuan paper, with the smallest water contact angle, has the largest surface energy. Regarding the different concentrations of soymilk-sized paper, the surface energy decreases with the increase in soymilk concentration. Previous studies have reported that the hydrophobic effect plays a significant role in protein structure stabilization and folding in soymilk [16]. The apolar part of soy protein establishes intramolecular contacts and releases water, while hydrogen bonds in the protein backbone contribute to the association and stabilization of the protein structure [17]. Among all the papers, the 1:1-sized sample, with the highest concentration of soymilk, exhibits an excellent folding performance of soy protein on the surface.

Figure 3 presents scanning electron microscopy (SEM) images of Sheng Xuan paper, both in top view and side view, along with the corresponding elemental mapping of soymilk-sized Xuan paper. It is evident from the images that a membrane-like coating is observable only on the soymilk-sized paper. Upon further magnification, a larger number of well-ordered papillae with an average diameter of 75 nm can be observed on the surface. The size distribution of papillae is mainly below 100 nm (inserted in Figure 3D), which is lower than the dynamic light scattering (DLS) result of soymilk, as shown in Figure 1C. Generally, the DLS technique provides the structure size of the aggregated state in liquid environments, which is larger than the SEM result. Conversely, the SEM images provide additional information about particle morphology and arrangement. Jiang’s work has previously illustrated that when a large volume of water droplets is placed on lotus leaf surface, it tends to flow out more easily through the margins of these papillae, rather than overflowing onto the upper surface [18]. When the water contact angle is high, it indicates that water has difficulty spreading across the paper’s surface, signifying lower water adhesion to the surface. As discussed above, the microstructured surface leads to a specific adhesion behavior, creating a high-energy barrier and contributing to the hydrophobic nature of the surface, with lower intrinsic surface energy. Furthermore, SEM-EDX analysis was employed to investigate the distribution of soy proteins within the paper. The SEM-derived elemental maps reveal an even distribution of C, N, O, and S elements, primarily originating from the soy protein. This observation further confirms that the soymilk permeates homogeneously into the paper fiber, ensuring a uniform distribution of soy protein throughout the paper structure.

The main component of paper is cellulose, which is a linear polymer consisting of linked D-glucopyranose units through β-1,4-glucosidic bonds. From a chain conformation perspective, cellulose can be viewed as an isotactic polymer of cellobiose [19]. The molecular structure of cellulose is depicted in Figure 4A. To investigate the structure of cellulose and its interaction with other molecules, nuclear magnetic resonance (NMR) techniques can be employed. Figure 4B,C present the ^1^H and ^13^C cross-polarization magic-angle spinning (CP/MAS) solid-state NMR spectra of Sheng Xuan and soymilk-sized paper. In the spectra, the chemical shift of 1.28 ppm can be attributed to protons in alkanes, while the peak at 29.55 ppm corresponds to methylene groups [20]. In addition, the proton peak at 4.5 ppm represents the hydroxyl proton at the C_3_ position of cellobiose, as well as proton peaks from the glucopyranose backbone and the amino proton. In the soymilk-sized paper, new peaks appear. Two peaks at 2.01 and 2.23 ppm are characteristic of acetyl methyl proton derived from soymilk, as indicated by the dotted line in Figure 4B. The methyl proton peak from soymilk is observed at 0.89 ppm [21]. These additional peaks confirm the presence of soymilk components in the sized paper. CP/MAS NMR is a valuable technique for analyzing hydrogen bonds in solid cellulose. In the ^1^H NMR spectra, a high-frequency shift at 26.28 ppm is observed in the soymilk-sized paper. The position of the hydrogen bond peaks in NMR spectra typically range from 8 to 20 ppm, while the peak at 26.28 ppm is significantly higher. However, previous studies have shown that the higher the strength of the hydrogen bond, the greater the peak shifts. This indicates the presence of stronger hydrogen bonds in the sized paper. In the ^13^C NMR spectra (Figure 4C), the C_1_ carbon peak of cellulose in Sheng Xuan paper appears at 104.8 ppm. In contrast, the C_1_ carbon peak in soymilk-sized paper shifts to a higher magnetic field, indicating the presence of intramolecular hydrogen bonds. The peaks assigned to C_2,3,5_ (70~80 ppm), C_4_ (80~90 ppm), and C_6_ (63.7 ppm) carbons remain unchanged [22]. The results obtained from the NMR spectra demonstrate a strong hydrogen-bonding interaction between soy protein and paper fibers. When a water-soluble protein undergoes folding into a compact and active conformation, the loss of polypeptide chain entropy is counteracted by favorable interactions within the protein and between the protein and its solvent. Non-covalent forces such as the hydrophobic effect, and hydrogen bonds, play significant roles in stabilizing the fold of the protein [16]. The soymilk-sized paper exhibits hydrophobicity and the presence of detected hydrogen bonds, both of which are essential for protein stability. These effects encourage the soluble protein to fold and remain in a stable state, which is supported by the observed surface energy characteristics.

### 2.3. Paper Properties during Accelerated Aging

To investigate the long-term behavior of soymilk-sized paper, paper samples were subjected to dry-heat aging conditions at the temperature of 105 °C. The presence of acids in the paper can catalyze the hydrolytic degradation of cellulose, which is the inherent instability of paper. Over time, the paper undergoes acidification due to natural aging processes. Consequently, the degree of polymerization decreases, resulting in a loss of paper strength. As shown in Figure 5A, all the soymilk-sized paper samples exhibit an alkaline pH initially, which is favorable for conservators. When nearly neutral soymilk is applied to Xuan paper, the paper’s fibers interact with the soymilk, leading to an initial increase in pH as the paper absorbs and interacts with the soymilk. However, the paper cellulose is ageing with the addition of soymilk. As the aging time progresses, the paper becomes progressively more acidic, resulting in a decrease in pH values. The most significant decline in pH occurs during the early stages of aging. The decrease in pH value is slower for paper samples with lower concentrations of soymilk. It is worth mentioning that the pH of soymilk-sized paper decreases at a much slower rate compared with Shu Xuan paper. The variation in pH among the soymilk-sized paper is influenced by the degradation and oxidation of the soy protein as well as the degradation of the paper cellulose. During the aging process, carboxylic acids, reactive oxygen species, and other products are generated from soymilk, promoting paper degradation. Hence, the pH values gradually decrease as the degradation progresses. Higher concentrations of soymilk result in faster degradation rates. As for the mechanical properties analysis of these various papers, it focuses on the folding endurance. In Figure 5B, the folding endurance of soymilk-sized paper is higher than that of Shu Xuan paper, and the folding resistance of all paper samples decrease with the prolongation of aging time. The trend is consistent with the observed pH values. The mechanical properties of the soymilk-sized paper exhibit improvements compared with the original Sheng Xuan paper, particularly during the aging process. Overall, the aging of soymilk-sized paper leads to acidification and a decline in pH values. However, the pH decrease is slower compared with traditional Shu Xuan paper. In addition, the soymilk-sized paper exhibits improved mechanical properties, particularly in terms of folding endurance, compared with the original Sheng Xuan paper.

Ms Qiu, the restorer, has observed that soymilk-sized paper exhibits a unique color that cannot be found in new paper. She specifically mentioned that the specified concentration of soymilk for sizing is one to three [6]. Figure 6 illustrates the chromaticity of different concentrations of soymilk-sized paper compared with the original Sheng Xuan paper. For all original paper samples, we compared the chromaticity changes with those of the Sheng Xuan paper. The results reveal that the 1:3 concentration of soymilk-sized paper exhibits the least chromaticity compared with the original Sheng Xuan paper. It is followed by 3:7, 1:1, and 1:5 soymilk-sized paper, and finally, the Shu Xuan paper. This finding suggests that the proposed concentration of 1:3 is reasonable and effective in replicating the color of historical paper artifacts during the restoration process. Furthermore, it is observed that the chromaticity of all the paper samples increases with the extension of aging time when compared with the corresponding original paper. The yellowing of cellulose during aging can be attributed to the presence of cellulose chromophores. The hydroxyl groups in the anhydroglucose units undergo transformations into carbonyl and carboxylic groups, resulting in the formation of aldehyde and ketone groups at positions C_2_ and C_3_. These changes contribute to the yellowing phenomenon of the paper [23]. Although soymilk sizing offers certain benefits, long-term preservation and careful monitoring are necessary to ensure the continued stability and longevity of the paper.

### 2.4. Lifespan Expectancy of Different Paper

To better maintain and preserve paper, the Ekenstam equation was utilized to predict the lifespan of these paper samples. The initial DP values, final DP values, and degradation rates at ambient temperature were required to calculate the lifespan values. Previous studies have shown that paper strength reaches zero around a DP value of 200 [24]. In this study, the final DP value was set to 200, and some empirical equations were employed, as described in the Section 3. Figure 7A shows the degradation rate at an aging temperature of 105 °C. The rate constants were determined from the slopes of linear fitting, based on Equation (6). The rate constants for Sheng Xuan, Shu Xuan, and 3:7, 1:3, and 1:5 soymilk-sized Xuan paper were found to be 4.79 × 10^−5^, 1.41 × 10^−4^, 9.42 × 10^−5^, 7.63 × 10^−5^, and 6.10 × 10^−5^ d^−1^, respectively. Subsequently, the degradation rate at ambient temperature was calculated using Equation (7). The pre-exponential factor in the equation is determined solely by the nature of paper degradation, independently of the reaction temperature and reactant concentration. Hence, the degradation constants at ambient temperature were calculated as 5.59 × 10^−9^, 1.64 × 10^−8^, 1.10 × 10^−8^, 8.92 × 10^−9^, and 7.21 × 10^−9^ d^−1^ for Sheng Xuan, Shu Xuan, and 3:7, 1:3, and 1:5 soymilk-sized Xuan paper, respectively. By reintroducing the final DP value of 200 into Equation (6), the lifespan of the paper samples was calculated and is depicted in Figure 7B. The results reveal that Sheng Xuan paper has the longest lifespan of 1385 years, while Shu Xuan paper has the shortest estimated lifespan of 396 years. This finding is consistent with previous conclusions, as the presence of alum in Shu Xuan paper promotes acid hydrolysis and facilitates paper degradation, creating an acidic environment within the pH range of 4.2 to 4.8, as observed in the pH characterization above [25]. For the different concentrations of soymilk-sized paper samples, the lifespan is negatively correlated with the concentration of soymilk. The longest estimated lifespan is 1153 years for 1:5 soymilk-sized paper, followed by 1:3 soymilk-sized paper with a lifespan of 987 years, and for 3:7 soymilk-sized paper, a lifespan of 680 years. However, an interesting phenomenon emerges where the folding endurance of 1:1 soymilk-sized paper surpasses that of 1:5 soymilk-sized paper after a month of accelerated aging. While soymilk may improve short-term properties like folding endurance, it may not provide sufficient protection against long-term degradation. The variation in lifespan among the soymilk-sized paper depends on the degradation and oxidation of the soy protein as well as the degradation of the paper cellulose. During the aging process, carboxylic acids, reactive oxygen species and other products are generated, promoting paper cellulose degradation [26,27,28]. In such cases, where the paper begins with an alkaline pH but gradually shift towards acidity over time, its alkaline reserve can be depleted. This depletion makes the paper more susceptible to the effects of aging. Higher concentrations of soymilk result in faster degradation rates. Nevertheless, it is important to note that these predictions provide a rough estimation of the usable life of the paper.

## 3. Materials and Methods

### 3.1. Materials

Commercially sourced soybeans were procured for this study. Soymilk extraction was carried out by a high-speed blender (WPB09J8, Westinghouse, Pittsburgh, PA, USA). Specifically, 600 mL water was added to 30 g soybeans to serve as an extraction solvent. The mixture was then processed in the high-speed blender for a duration of 6 min at room temperature. Subsequently, the raw soymilk was filtered through a cotton filter to eliminate the majority of insoluble impurities present in the soymilk. The resulting solution was then employed for further applications and subsequent analyses.

Diiodomethane was acquired from Shanghai Aladdin Biochemical Technology Co., Ltd., Shanghai, China, while Bis(ethylenediamine)copper(II) hydroxide (1 M in H_2_O, or cupri-ethylenediamine) was supplied by Energy Chemical, Anhui Zesheng Technology Co., Ltd., Hefei, China, All chemicals were used as received without requiring additional purification. Furthermore, all solutions utilized throughout the experiment were prepared with deionized water. The initial handmade Xuan paper samples employed in this study were purchased from Hongxing Xuan Paper Co., Ltd. (Xuancheng, China). Specifically, writing-grade handmade Xuan paper composed of bark fiber from *Pteroceltis tatarinowii* and straw fiber was utilized. It is worth noting that Xuan paper can be categorized into various styles based on the extent of sizing. In this study, Sheng Xuan (unsized) paper and Shu Xuan (sized with alum–gelatin solution) paper were utilized as controls.

### 3.2. pH, Viscosity, Particle Size, and Zeta Potential of Soymilk

The stock soymilk was prepared following the procedures outlined in Section 3.1 above. To obtain different concentrations of soymilk, various volume ratios of the stock soymilk and deionized water were combined in 10 mL centrifuge tubes by pipettes. The pH values of the different soymilk concentrations were measured with a pH meter (FE28, Mettler–Toledo International Inc., Shanghai, China). Viscosity measurements of the different soymilk concentrations were conducted at a temperature of 20 °C using a Digital Display Rotary Viscometer (NDJ-8S, Shanghai Lichen Bangxi Instrument Technology Co., Ltd., Shanghai, China).

The viscometer was equipped with different types of concentric cylinder geometry rotors. The surface charge and particle size of soymilk at various pH values were determined via a Zetasizer Nano ZS90 Analyzer. These measurements were conducted in triplicate to ensure accuracy. The pH values were adjusted within the range of 3.64 to 8.54 using a tiny amount of HNO_3_ or NaOH solutions. All measurements were performed at ambient temperature.

### 3.3. Basic Weight Measurement

To facilitate convenient sizing degree quantification, a paper cutter was employed to obtain circular samples with a diameter of 125 mm. Subsequently, the Sheng Xuan paper was immersed in various concentrations of soymilks to produce different types of soymilk-sized Xuan paper. After the sizing treatment, the paper was dried and stored under constant humidity and temperature conditions (humidity of 60 ± 2%, 25 ± 1 °C). The basic weight of the paper was calculated following the ISO 534:2011 standard [29]. In essence, the basic weight of the paper is determined by dividing the weight of the paper by its surface area.

### 3.4. Static Contact Angles and Surface Energy of Different Types of Paper

The static contact angles of water, observed from a side view, were measured with the OCA-20 contact-angle system (Dataphysics Instruments GmbH, Filderstadt, Germany) at room temperature. To determine the surface energy of the soymilk-sized paper, Owens theory was applied, which involves calculating the surface energy by two components: apolar and polar [30]. By measuring the contact angles of water and diiodomethane on the paper surface, the apolar and polar values of the paper’s surface energy were obtained. The total surface energy of the paper is approximately the sum of these two components. It is known that the total surface tensions of water and diiodomethane are 72.8 and 50.8 mJ/m^2^, respectively. The corresponding apolar parts are 21.8 and 48.5 mJ/m^2^, while the polar parts are 51 and 2.3 mJ/m^2^, respectively. By substituting the data of the surface tension and contact angle of water and diiodomethane into the equation below, two dependent equations can be derived. These equations allow for the calculation of the surface energy of the soymilk-sized paper.
(1)$$\gamma_{L}\left(1+cos\theta \right)=2{\left(\gamma_{S}^{a}\cdot \gamma_{L}^{a}\right)}^{1/2}+2{\left(\gamma_{S}^{p}\cdot \gamma_{L}^{p}\right)}^{1/2}$$

Herein, $\gamma_{L}$, $\gamma_{L}^{a}$, and $\gamma_{L}^{p}$ represent the total, apolar, and polar surface tension of detecting liquid, respectively. The parameter θ is the static contact angle of the specific detecting liquid on the paper. $\gamma_{S}^{a}$ and $\gamma_{S}^{p}$ represent the apolar and polar parts of the solid surface energy that need to be determined.

### 3.5. Morphology Analysis of Different Types of Paper

A small quantity of each sample was affixed to conductive adhesive, enabling the investigation of the morphologies and distribution of elements on both the front and side surfaces of the paper using a HITACHI SU8010 scanning electron microscope (Hitachi High-Technologies corporation, Tokyo, Japan). Since the sample is non-conductive, it was necessary to apply a thin layer of gold onto the surface of the sample. This gold coating, consisting of several to a dozen atomic layers, is only a few nanometers to a dozen nanometers thick. Importantly, this gold coating has minimal impact on the appearance of the sample. Micrographs at various magnifications were captured at the energy of 10 KeV.

### 3.6. Solid-State Nuclear Magnetic Resonance Analysis

Due to the paper samples’ insolubility in the typical solvents used for NMR analysis, the assessment of the paper’s chemical structure was conducted using solid-state nuclear magnetic resonance (ssNMR). The spectra were acquired using a Bruker Avance II 400 MHz spectrometer (Bruker BioSpin GmbH, Ettlingen, Germany) operating at 400.18 MHz (^1^H) and 100.62 MHz (^13^C). Powdered samples were placed in a 4 mm rotor and spun at a rate of 10 kHz using a double air-bearing probe head. To obtain reliable results, each sample was subjected to 128 scans during the ssNMR analysis. The ^13^C chemical shifts were referenced to adamantane, which serves as a standard reference compound in ssNMR spectroscopy.

### 3.7. Color Analysis of Different Types of Paper

The chromaticity of the paper was determined with an automatic colorimeter (ZB-A, Hangzhou Zhibang Automation Technology Co., Ltd., Hangzhou, China). To ensure accuracy and minimize potential errors, each sample was measured six times at different locations. The change in chromaticity ∆E was calculated via the following equation:(2)$$∆E=\sqrt{∆L^{2}+∆a^{2}+∆b^{2}}$$

In the equation, *L*, *a*, and *b* represent three different colorimetric coordinates values, respectively. ∆L, ∆a, and ∆b are the differences between corresponding values of different samples.

### 3.8. pH Measurement and Mechanical Performance Testing

The measurement of paper pH differed from that of soymilk. In this study, a pH meter (pH-100B) provided by Shanghai Lichen Instrument Technology Co., Ltd. (Shanghai, China) was used. The measurement procedure followed the ISO 6588-1:2021 standard [31]. A drop of water was put on the paper surface, and the electrode head was vertically immersed in the water droplet. The pH value was recorded after allowing a 5 min equilibrium time. Each sample was tested five times at different locations, and the measurements were conducted at room temperature.

The mechanical performance of the paper was assessed by the folding endurance index, following the ISO 5626:1993 standard [32]. The paper specimen was subjected to longitudinal tension and folded forward and backward until it broke. The folding resistance was determined as the logarithm of the number of double folds before the specimen fractured. In this context, double folding refers to reciprocating the sample back and forth along the same fold line. The folding resistance is expressed as the logarithm of the number of double folds until the sample fractures. The MIT folding method was employed, and an MIT Folding Endurance Tester (Model: ZB-NZ135A) supplied by Hangzhou Zhibang Automation Technology Co., Ltd. (Hangzhou, China) was used for the folding endurance test.

### 3.9. Lifespan Evaluation of Different Types of Paper

The degree of polymerization (DP) of cellulose plays a crucial role in determining its physical and chemical properties, making it an important factor to consider when working with paper. The DP values of paper cellulose were measured following the ISO 5351:2010 standard [33]. The measurement procedure involved weighing a specific amount of paper and adding it to a 50 mL centrifuge tube along with 10 mL of deionized water. After allowing the paper fibers to swell, a 10 mL cupri-ethylenediamine solution was added, and the paper specimen was completely dissolved. The resulting solution was then transferred to an Ubbelodhe viscometer, and the specific viscosity ($\eta_{sp}$) of the dissolved solution relative to the cupri-ethylenediamine solvent was recorded. Importantly, the temperature was maintained at 25 °C throughout the measurement. After obtaining the specific viscosity, the intrinsic viscosity ($\left[\eta \right]$) was calculated via the Martin empirical equation below [34]:(3)$$\eta_{sp}=\left[\eta \right]\cdot \rho \cdot e^{K\left[\eta \right]\rho }$$

Herein, ρ represents the concentration of paper cellulose (in g/mL), and K is an empirical constant specific to the cellulose–copper ethylenediamine system, equal to 0.13. Subsequently, the molecular weight (M) of cellulose, which is usually expressed as the DP, was determined via the Mark–Houwink equation:(4)$$\left[\eta \right]=KM^{\alpha }$$

In this equation, K and α are constants specific to the system. Hence, the Mark–Houwink equation can be rewritten as follows:(5)$$DP^{0.905}=0.75\left[\eta \right]$$

The paper’s life expectancy can be predicted via the Ekenstam equation:(6)$$\frac{1}{DP_{n}}-\frac{1}{DP_{0}}=kt$$
where $DP_{n}$ and $DP_{0}$ are the paper’s degree of polymerization at any given moment and the initial moment, respectively. k (day^−1^) is the rate constant of degradation, and t is the accelerated aging time of the paper sample. Since the accelerated aging experiment was conducted at 105 °C, the obtained degradation rate constant should be converted to the corresponding rate constant at 25 °C. This conversion can be accomplished via the Arrhenius equation:(7)$$lnk=lnA-\frac{E_{a}}{RT}$$
where A represents the pre-exponential factor, which is temperature-independent. $E_{a}$ denotes the activation energy, which has been experimentally determined to be 106 kJ/mol. R is the gas constant, and T is the absolute temperature [35]. By obtaining the degradation rate constant, defining the DP limit as 200, and plugging the relevant values into the Ekenstam equation, the lifespan of the paper can be evaluated [36].

## 4. Conclusions

This study focused on the application of soymilk as a sizing agent for Xuan paper and its effects on various properties and long-term behaviors, especially for the potential mechanism of sizing paper with soymilk. The soymilk-sized paper exhibited hydrophobicity, improved mechanical properties, and a unique chromaticity that was not found in new paper. The microstructure, protein folding, and hydrogen-bonding interactions between soy protein and paper fibers contributed to these properties. Moreover, the soymilk-sized paper showed resistance to acidification, a slower pH decrease, and decreased basic weight compared with traditional alum–gelatin-solution-coated paper. The aging process led to acidification, decreased mechanical strength, increased chromaticity, and degradation of the paper samples. Predictions using the Ekenstam equation estimated the lifespan of the paper, with Sheng Xuan paper having the longest estimated lifespan and soymilk-sized paper exhibiting negative correlations between lifespan and soymilk concentration. These findings provide valuable insights for the preservation and long-term maintenance of paper using soymilk sizing, in spite of presenting a rough estimation, and further preservation strategies should be considered for optimal paper conservation.

## Figures and Tables

**Figure 1 molecules-28-06791-f001:**
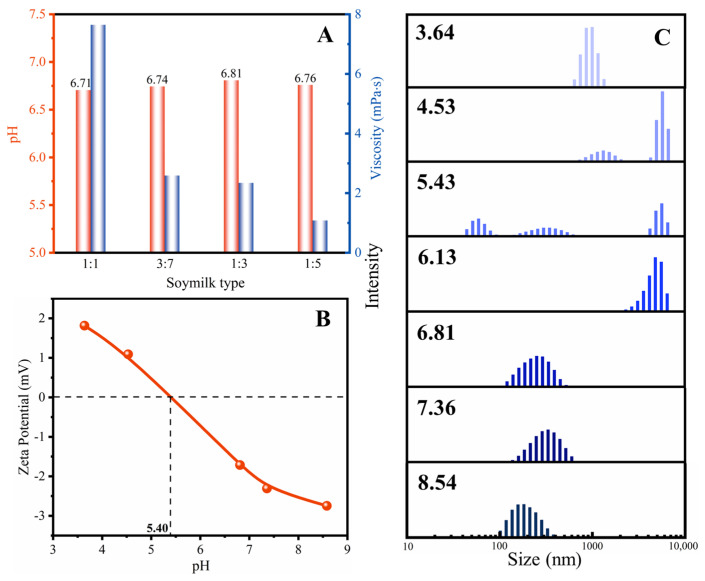
(**A**) pH and viscosity of different concentrations of soymilk, (**B**) zeta potential, and (**C**) size distribution of soymilk at various pH values.

**Figure 2 molecules-28-06791-f002:**
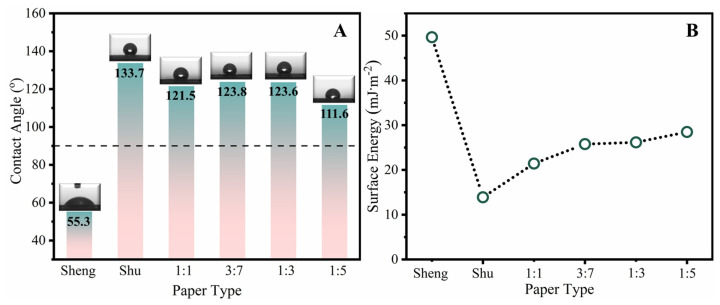
(**A**) Static contact angle and (**B**) surface energy of different papers.

**Figure 3 molecules-28-06791-f003:**
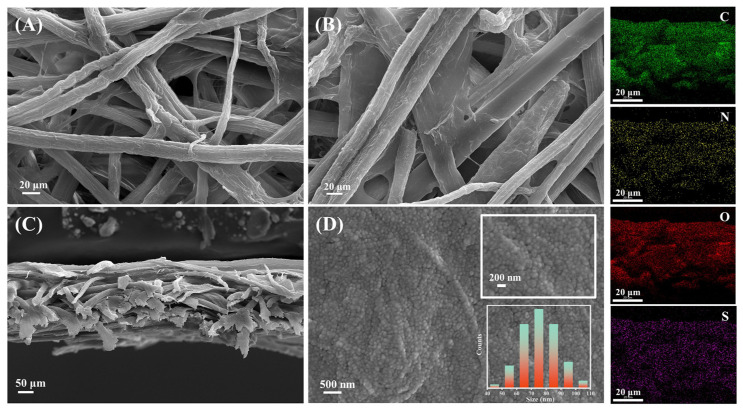
SEM images of (**A**) Sheng Xuan; soymilk-coated Xuan paper from the top (**B**) and side (**C**) view; and (**D**) the microstructure of soymilk-sized paper and corresponding elemental mapping.

**Figure 4 molecules-28-06791-f004:**
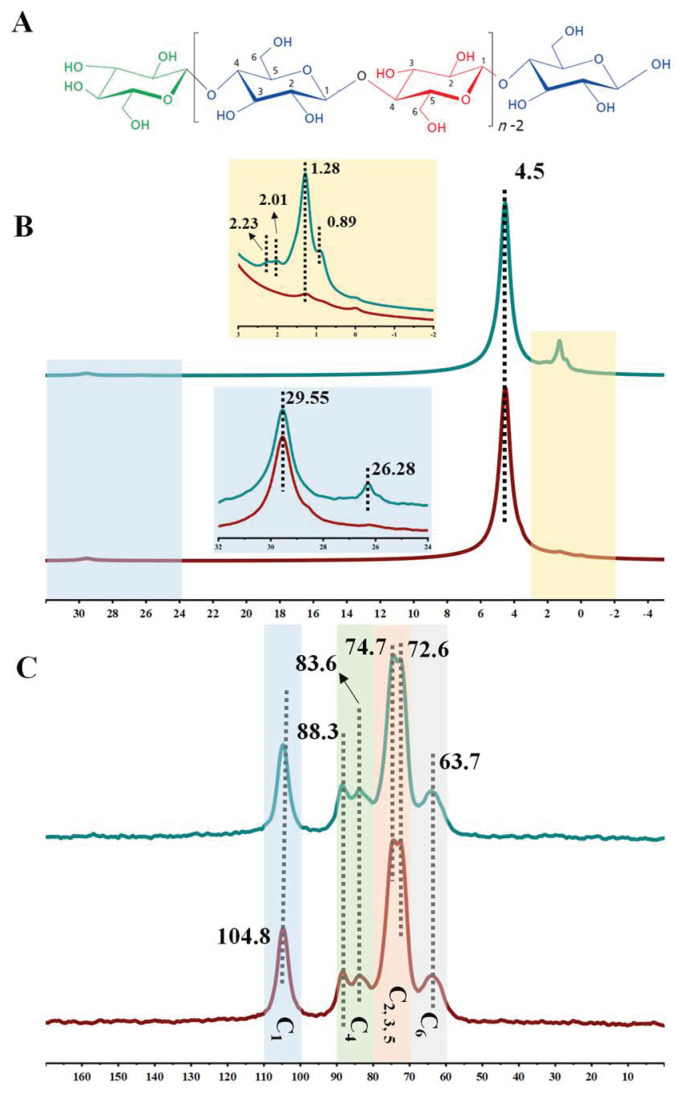
(**A**) The molecular structure of cellulose; (**B**) ^1^H MAS NMR spectra and (**C**) ^13^C CP/MAS spectra of Sheng Xuan paper (red, lower) and soymilk-sized paper (green, upper).

**Figure 5 molecules-28-06791-f005:**
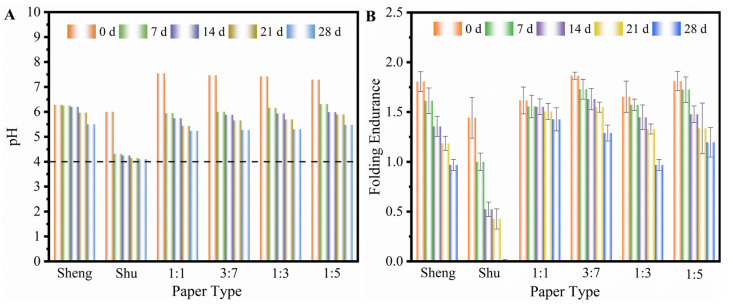
(**A**) The pH values and (**B**) folding endurance of different papers at various time stages.

**Figure 6 molecules-28-06791-f006:**
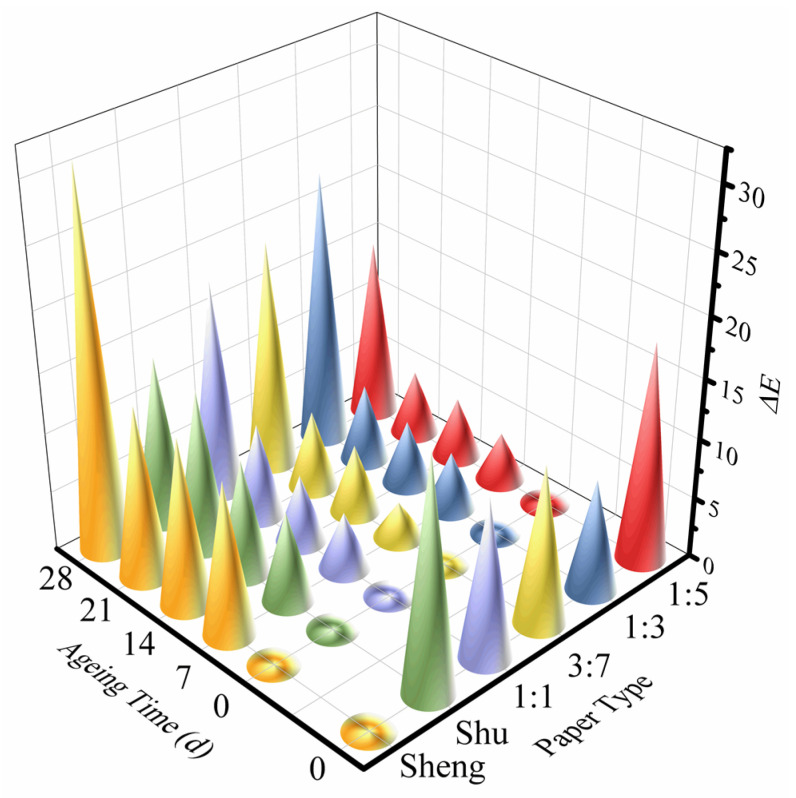
The chromaticity of different papers at various time stages.

**Figure 7 molecules-28-06791-f007:**
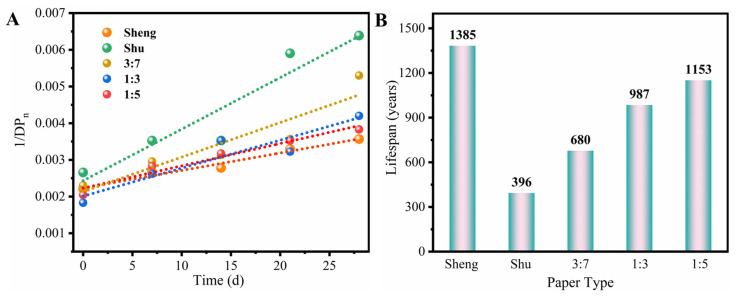
(**A**) Rate constants of different papers under accelerated aging conditions; (**B**) lifespan of different papers calculated using Ekenstam equation.

**Table 1 molecules-28-06791-t001:** Basic weights of different papers.

Paper Type	Basic Weight (g/m^2^)
Sheng	21.31
Shu	30.44
1:1	27.94
1:3	25.98
3:7	24.70
1:5	22.55

## Data Availability

The authors confirm that the data supporting the findings of this study are available within the article.

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
