# Peer review of "An Investigation into the Performance and Mechanisms of Soymilk-Sized Handmade Xuan Paper at Different Concentrations of Soymilk"

_molecules, 2023, doi:10.3390/molecules28196791_

Round 1

Reviewer 1 Report

Authors have studied application of soy-milk as sizing agent for Xuan paper. The work is very interesting. However, the size determination part is not very clear. In case of size determination of soy milk, figure 1 (c) does not give a clear picture of size. A better DLS size bar graph having clear size scale is recommended. Authors may include a separate section on DLS characterization and provide more detailed information on size and size distribution, along with zeta potential.

Similarly, in case of size determination of soymilk coated Xuan paper, enough information on size and surface morphology is not provided from SEM analysis (figure 3). The SEM images look great with unique morphology, however, no discussion is included in the manuscript. Moreover, the image quality is poor when zooming into 200 nm scale. This specific SEM image in 200 nm scale should be replaced with a better quality image having proper size scale bar for average size calculation. Usage of some image software e.g. imageJ would be helpful.

Lastly, DLS experiment should be performed on soy milk sized Xuan paper, in order to provide information on change in size and size distribution of soymilk coated paper. DLS provides hydrodynamic size while SEM provides actual size. Therefore, DLS size of soymilk and SEM size of soymilk coated paper is not comparable.

The authors have used unnecessarily long and complex sentences throughout the manuscript. Splitting long and complex sentences into short and simple sentences with proper scientific soundness is recommended. This is necessary for better interest to the readers.

Reviewer 2 Report

The authors investigated the properties of soymilk sized Xuan paper such as surface density, water contact angle, folding endurance, etc. varying concentration of soymilk. The experiments were systematically performed and summarized well. But, I found some concerns need to be addressed.

1. Please briefly explain the difference of Sheng Xuan paper and Shu Xuan paper.

2. Why do you mention polysaccharide-protein complex in line 36? Is there polysaccharide in soymilk?

3. In line 160-170, the authors use similar but different terms such as surface density, basic weight, and paper density. Is theses terms different? If not, please use one term.

4. Are the papillae due to the coating of soy protein?

5. Why are you saying that the microstructured surface specific adhesion? Hydrophobic and high water contact angle means lower adhesion of water to the surface.   

6. Why the soymilk-sized paper is initially alkaline?

7. It seems 1:1 paper have higher amount of soymilk coating based on higher surface density. Then it seems the pH should decrease more slowly. But decrease of pH slower for 1:5 paper. Why this happens?

8. In Figure 6. Some data are compared to the Sheng Xuan paper, and some data are compared to the original paper. Please give information in Figure 6 as well as in text.

9. So, the lifespan estimated from DP is longer for 1:5 paper. But, according to Figure 5B, the lifespan from folding endurance is longer for 1:1 paper. What is the reason do you think? It seems that lifespan estimated from DP is the lifespan of soymilk coating itself rather than soymilk sized paper. Please give your discussion.

10. Title of 3.1 is wrong (line 331).

Reviewer 3 Report

The manuscrint 2585883Investigation into the Performance and Mechanism of Different Concentration Soymilk Sizing on Handmade Xuan Paper” is described the method for conservation of the paper with soymilk.

Some remarks on the text are listed below.

1.     Graphical abstract is hardly understood: numbers 1 and 2 are not depicted, dashed lines describing the interaction of amino acid with carbohydrates are not correct, only two colors of several in a structure of modified paper are decoded.  

2.     Figure 1. (A) pH and viscosity of different concentrations of soymilk, (B) Zeta potential, and (C) size distribution of soymilk at various pH values. Distribution of what particles of soymilk?

Table 1. What paper, Sheng or Shu, is sized with milk?

Figure 3. SEM images and elemental mapping of soymilk-coated Xuan paper – C, N, O, S, especially N and S, are not visible well.

Figure 6. Two lines are noted by “0” in the coordinate “Ageing time”, and the difference is not described.

Round 2

Reviewer 1 Report

The authors have successfully answered to most of the reviewer's concerns. The manuscript can now be accepted in its present form.

Overall english looks fine. Minor checking is required.

Reviewer 2 Report

The manuscript was successfully revised according to the comments.